# Participants' perspectives on a multimodal stress management and comprehensive lifestyle modification program for patients with Crohn's disease—A qualitative interview study

**Christoph Schlee**[1,2,3]**, Christine Uecker**[1,2]**, Özlem Öznur**[1,2]**, Nina Bauer**[1,2]**, Jost Langhorst**[1,2] *

1 Department of Internal and Integrative Medicine, Sozialstiftung Bamberg, Bamberg, Germany,
2 Department of Integrative Medicine, Medical Faculty, University of Duisburg-Essen, Bamberg, Germany,
3 Institute for Sociology, University of Bamberg, Bamberg, Germany

* jost.langhorst@sozialstiftung-bamberg.de

**Data Availability Statement:** Data cannot be shared publicly because of data protection and ethical reasons. Anonymized data are available

## Abstract

### Background

Crohn's disease (CD) is a type of inflammatory bowel disease (IBD) that is prevalent world-wide and associated with reduced quality of life for patients. Multimodal therapy approaches, which emphasize lifestyle modifications such as mindfulness and stress reduction, can be promising in enhancing health-related quality of life for IBD patients. However, research on multimodal therapy approaches for CD remains insufficient.

### Method

This qualitative interview study is part of a mixed-methods approach that is embedded in a randomized controlled trial. It investigates the impact of a comprehensive 10-week day clinic lifestyle modification program on the health condition and quality of life of CD patients. Telephone interviews (n = 19) were conducted three months after the program to examine individuals' viewpoints on the intervention, including perceived changes and transfer of elements into daily life. Reflexive thematic analysis was performed using MAXQDA software.

### Results

The results indicate that CD can have very individual and comprehensive impacts (psychological, physical, social), leading to reduced perceived quality of life and well-being. By participating in the program, patients wanted to find self-help options to complement conventional pharmacotherapy and actively manage their disease. Patients expressed high satisfaction with the program, feeling it provided valuable support for daily disease management. They were able to integrate adequate therapy elements into their routines to

from the Department of Internal and Integrative Medicine for researchers who meet the criteria for access to confidential data. Contact: Email: FIGN@sozialstiftung-bamberg.de Sozialstiftung Bamberg Klinik für integrative Medizin und Naturheilkunde Buger Straße 80 96049 Bamberg Germany.

**Funding:** This study as part of the project "Integrative Medizin in Bayern 2020 (IM-BAY 2020)" was supported by the Bavarian State Ministry for Health and Care (Germany) by means of the funding program Gesund.Leben.Bayern, https://www.stmgp.bayern.de/ Reference Number GE7-2497-GLB-19-V4. The funders had no role in study design, data collection and analysis, decision to publish, or preparation of the manuscript.

**Competing interests:** I have read the journal's policy and the authors of this manuscript have the following competing interests: JL was a speaker for Repha GmbH, Techlab Inc., Falk Foundation, Takeda, Celegene GmbH and Willmar Schwabe and received research funding from Repha GmbH, Techlab Inc., Falk Foundation and Willmar Schwabe. The sponsors had no role in the design and execution of the study, interpretation of the results or writing of the manuscript. The remaining authors declared that the research was conducted in the absence of any commercial or financial relationships that could be construed as a potential conflict of interest.

complement their care. Patients recognized significant improvements in various domains, mainly in the psychological domain, e.g., improved self-efficacy, symptom management, and, also partly physical/symptomatic and social improvements.

## Conclusion

A multimodal stress reduction and lifestyle modification day clinic appears to be beneficial as a complementary therapy for CD patients. It offers additional options and helps patients to address individual symptoms and needs, improve their understanding of the disease and their quality of life. Although promising, further research is needed to assess its long-term effects.

## Trial registration

ClinicalTrials.gov, identifier: NCT05182645.

## Introduction

Worldwide, approximately 6.8 million people suffer from inflammatory bowel disease (IBD), more than 2 million of whom are European [1]. IBD is a chronic relapsing inflammatory disorder of the gastrointestinal tract [2, 3] with the main types Crohn's disease (CD) and ulcerative colitis [2]. The incidence of CD was increasing in many regions. Although IBD is especially prevalent in western industrialized countries, the incidence of IBD cases is increasing worldwide, especially in newly industrialized countries [4]. Overall, it is evident that IBD patients suffer from a reduced quality of life [1], due to symptoms, such as abdominal pain, diarrhea, flatulence, as well as psychosocial symptoms. The causes and factors influencing CD are not yet clearly understood [5, 6]. However, it is assumed that, in addition to genetic factors, environmental factors, or lifestyle factors such as perceived stress in everyday life play a major role and impact the course of the disease [3, 7, 8]. Poor dietary habits and deficiencies in some nutrients are also implicated in the etiology of IBD [9–11]. In addition, it is suggested that the predominant dietary habits, e.g., highly processed foods, sugar consumption, etc., in modern societies can have a negative influence [12–14].

Initial studies indicate that multimodal approaches that address lifestyle modification of patients with IBD can be used promisingly in prevention, but also in complementary therapy [7, 15, 16]. Mind-body therapies, mindfulness, meditation, relaxation, stress management programs, and yoga can improve the impaired quality of life of IBD patients and, in some cases, relieve pain [8, 17–26]. In patients with ulcerative colitis, our research team has already shown that a comprehensive program of stress reduction and lifestyle modification (including, inter alia, nutrition therapy and phytotherapy) in the form of a day clinic, which is also the approach used in the present study, can improve quality of life, as perceived stress can be reduced and, in parts, also physical symptoms [16, 27, 28].

In addition to the general lack of studies on multimodal therapy concepts for CD [29, 30], there is a lack of insight into the subjective experiences with mindfulness interventions in IBD from the perspective of the participating patients [28, 31]. This insight could be of crucial importance to better understand the complexity of the disease and the need for individual therapy from the patient's perspective [32]. In addition, patient experience can be used to

continuously improve the day-clinic intervention, to provide therapy tailored to patient needs [28], and thus to ensure the appropriate use of health care resources.

This qualitative interview study presented here is nested in a randomized controlled trial [29, 30] which aimed to investigate the effects of participation in a multimodal stress reduction and lifestyle modification program on the quality of life of patients with CD. The qualitative study approach aimed to provide an additional perspective on the intervention, to complement previous quantitative research interests, as well as to develop new insights. Specifically, the following research questions were addressed: How do patients with CD describe their everyday life? Why did the patients decide to participate in the day clinic program? How did patients experience the day clinic program? Do patients perceive changes regarding quality of life and CD symptoms due to the intervention? How could patients integrate program elements into their daily lives?

## Materials and methods

### Study design and day clinic program

The study was conducted as a mixed methods approach (see Fig 1) at the Klinikum Bamberg (Bamberg, Germany) in 2020 and 2021 –the recruitment and data collection process began on 01.09.2020 and ended on 14.12.2021 –that examined the effects of a mind-body medicine and comprehensive lifestyle modification training program (10-week day clinic program) on quality of life in patients with CD [29, 30]. The study was approved by the Ethics Committee of the Bavarian Medical Association (BLÄK, No. 19096), registered at clinicalTrials.gov (NCT051826745), and conducted according to the Declaration of Helsinki and good clinical practice guidelines.

This qualitative study served as a supplementary study to the randomized controlled trial. The quantitative research results, for instance the feasibility of the study [29] and the pre-post and follow-up study results of the randomized controlled trial [30] have already been published. This paper focuses on the qualitative research interest, see Fig 1. To deepen, complement, and extend the quantitative findings of the study, post-intervention interviews were conducted with patients from the intervention group three months after completion of day clinic program. Due to the COVID-19 pandemic it was decided to reduce the contacts to a minimum and thus implement interviews via telephone and not face-to-face. The aim was to gain a qualitative perspective i.e., to explore individual perceptions of the intervention by the patients interviewed (see again Fig 1) [30]. More specifically, the following topics were examined: experience of disease in daily life, reasons for participating in the day clinic program, experiences with the program, perceived changes due to the intervention, and integration of program elements in daily life.

The mind-body-medicine-based multimodal stress management and comprehensive lifestyle modification training program (see S1 Fig) consisted of 10 consecutive weekly group sessions (6 hours per weekly session) from 11 am to 5 pm. It was led by an experienced mind-body instructor. The intervention groups consisted of five to seven participants who attended the sessions together. The participants received theoretical lessons and practical training. In this program, topics such as mindfulness-based stress reduction (e.g., relaxation, body scan), naturopathic self-help strategies (e.g., hydrotherapy/water applications according to Kneipp, wraps and pads), herbal remedies for gastrointestinal symptom control, exercise therapy (e.g., walking), yoga, qigong, and nutritional therapy including cooking classes based on a Mediterranean whole foods diet were essential. The participants were provided with audio guides of relaxation and mindfulness-based exercises and a variety of print information material for home use trial [29, 30].

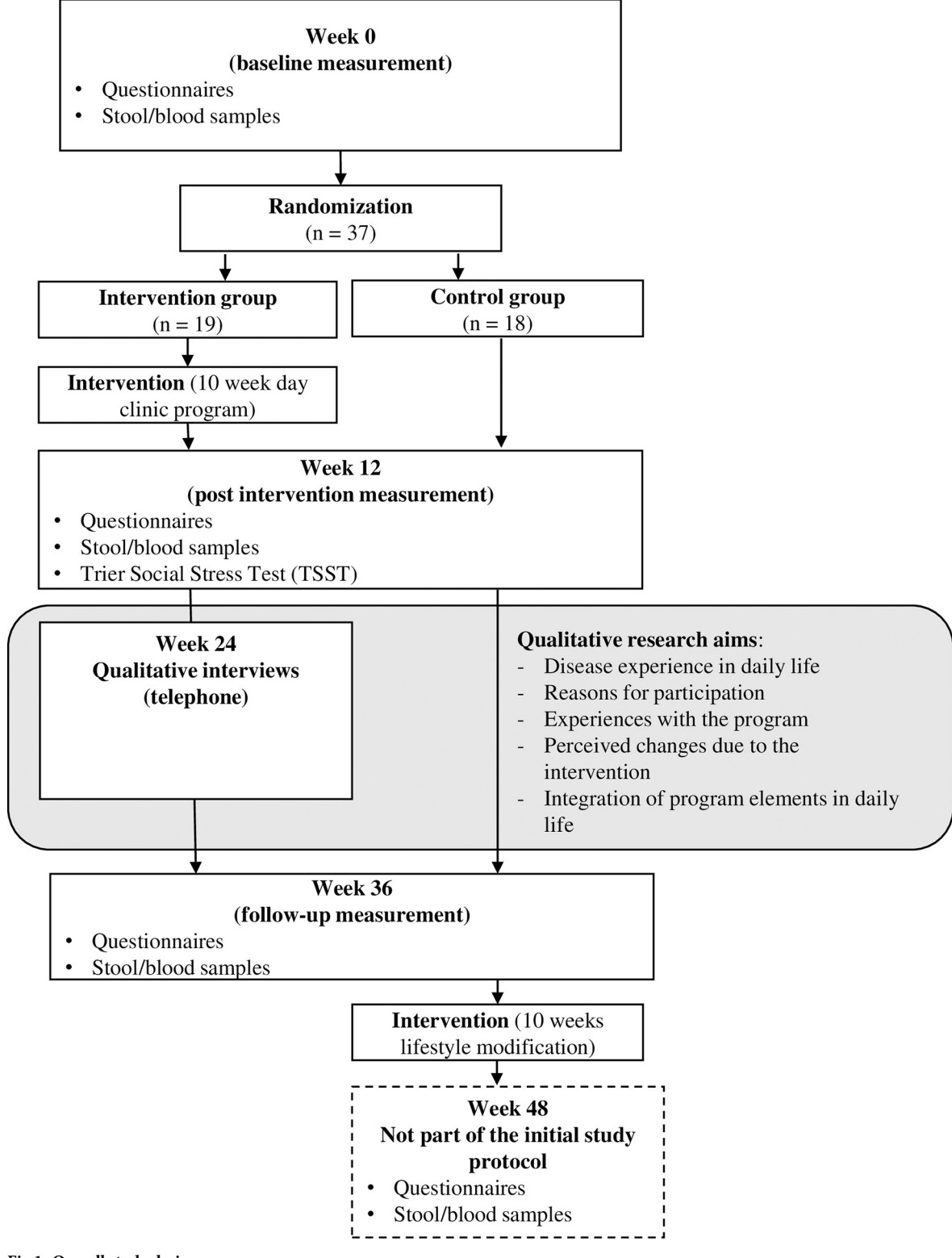

**Fig 1. Overall study design.**

## Participants

A total of 37 patients diagnosed with CD were enrolled in the randomized controlled trial. In this respect no sample size calculation was done because this was a feasibility and pilot study. Inclusion criteria were age 18 to 75 years, confirmed diagnosis of CD, stable medication for at least three months, and signed written informed consent. Not eligible to participate in the study were patients with highly acute course, complete colectomy, severe mental illness, severe somatic comorbidity, pregnancy, and participation in stress-reduction programs or clinical trials involving psychological intervention during the study period. Participants were randomly assigned to the intervention and control groups [29, 30]. 19 patients were in the intervention and 18 in the control group. Members of the intervention group participated in the program for 10 weeks and were given informational material in the form of a booklet which contains general information about the disease, mind-body medicine and self-help strategies. The control group did not participate in the program right away but attended a single 90-minute education session with information for self-directed application. Participants randomized to the control group had the option to attend the day-clinic program after the waiting period and follow up assessment.

The sample for this qualitative study (see Table 1) consisted of all 19 patients ($n$ = 19) from intervention group who varied in terms of sex, age in years, time since initial diagnosis in years, disease activity, quality of life. Thus, it was possible to cover different subjective perspectives and experiences, as well as to identify relevant patterns across the interviews [33, 34].

## Semi-structured interviews

To have the opportunity to cover specific topics in the interviews as well as to consider prior theoretical considerations and knowledge, a semi-structured approach using an interview guideline was chosen [35]. Furthermore, this qualitative approach allows for previously unexpected insights and enables participants to contribute their own topics and opinions [36, 37]. The interview guideline was primarily developed deductively, included open-ended questions shown in Table 2, and was orientated on previous study on patients with ulcerative colitis of our research team [28]. The average duration of an interview was approximately 40 minutes. They were audio-recorded following patient consent, transcribed verbatim, and anonymized [38]. After the interviews, the interviewers completed interview protocols as field notes to document the interview situation and any peculiarities or problems that arose during the interviews. In this respect, there were no particular incidents or problems that would have affected

**Table 1. Description of the qualitative sample (intervention group).**

|  |  | intervention ($n$ = 19) | control ($n$ = 18) |
|---|---|---|---|
| **Sociodemographic characteristics** |  |  |  |
|  | sex |  |  |
|  | male | **6 (32%)** | 6 (33%) |
|  | female | **13 (68%)** | 12 (67%) |
|  | age (years) | **49.6 (13.1)** | 46.8 (11.4) |
| **Disease related factors** |  |  |  |
|  | Years since initial diagnosis | **19.2 (11.6)** | 21.8 (10.6) |
|  | Disease activity (HBI) | **6.2 (3.8)** | 4.8 (2.4) |
|  | Quality of Life (IBDQ) | **139.9 (27.3)** | 155.3 (25.2) |

*Note.* This table shows the randomized controlled trial sample, where the intervention group represents the interview study participants. HBI: no disease activity = 0, highest disease activity = 30; IBDQ: lowest IBD related QOL = 32; highest IBD related QOL = 224, relative frequencies/standard deviation in brackets [30].

**Table 2. Main topics, research aims, and exemplary questions from the interview guideline.**

| Main topics and research aims | Exemplary questions and narrative stimuli |
|---|---|
| ➢ Experiences with the disease in everyday life and applied measures to counteract it | Please tell me about your personal experiences with Crohn's disease in everyday life.<br>What have you done to treat your disease (the effects of the disease) so far? |
| ➢ Motivation for participating in the day clinic program and expectations | Why did you decide to participate in the program?<br>What were your expectations in advance? |
| ➢ Perceived effects of participation<br> • Changes in everyday life<br> • Perception of the effectiveness on or change of the disease/one's life | Can you please tell me if you have noticed any changes compared to before the program?<br>Did/does the program affect your life? The disease? If so, in what way? How does it manifest itself? |
| ➢ Integration of elements of the program<br> • Implementation of techniques and program elements<br> • Useful aspects and problems/hurdles<br> • Future use of techniques and program elements | How did/do you integrate contents/elements of the program into your everyday life? Which techniques do you use most, which ones less? Why?<br>What was helpful in implementing the program in your everyday life? Any obstacles?<br>How likely do you think it is that you will use the techniques in the future? |
| ➢ Perception of the program<br> • Satisfaction<br> • Positive and negative aspects<br> • Suggestions for improvement | How did you like the program?<br>What did you find particularly good? What did you find less good?<br>What do you think should have been different and what else do you need for yourself? |

the quality of the interview. Two sociologists and one person with a Master's degree in naturopathy and complementary medicine were the interviewers in this study. Each interview was conducted by a single interviewer. All interviewers are experts in conducting qualitative interviews. To ensure maximum openness in conducting the interviews and to avoid any influence on the interview situation, the interviewers were not personally known to the participants, had no previous contact with the participants, and were not involved in the intervention.

## Data analysis

Reflexive thematic analysis was used to identify relevant themes and patterns in subjective meanings of the participants. Interpretation and hermeneutic procedures enabled to access latent meaning in addition to semantic meaning [34, 39]. This approach is located in the field of symbolic interactionism, since the interest in knowledge lies in the subjective views and life circumstances of the patients and data was obtained from communication [40]. A combination of inductive and deductive elements, i.e., data-driven and theory-driven coding, was chosen. Therefore, the categories based on the empirical data as well as on the specific research questions and theoretical considerations. In part, considerations were based on previous research of day clinic intervention in patients with ulcerative colitis [28]. The interdisciplinary research team from psychology, sociology, biology, and medicine used the software MAXQDA for coding processes. All participants of the intervention group were interviewed and included in the analysis. Thus, no purposeful sub-sampling [33] was done. During the process of analysis, the procedure and the results were discussed and reflected in the research team to ensure validity, transparency and intersubjectivity [41, 42].

## Results

### Comprehensive and multiple impacts of Crohn's disease on patients' everyday life and well-being

The analyses reveal that for the majority of patients, the disease is accompanied by moderate to severe *physical consequences* (symptoms) such as diarrhea, abdominal pain, frequent bowel movements, etc. On the *psychological level* these circumstances were perceived as stressful and burdening for the patients. In the *social context* the symptoms appear problematic as well, i.e.,

in interaction with other people, especially when the patients are outside their own homes, e.g., at work or for leisure time activities. They also report personal restrictions or lack of understanding on the part of other people. In addition, there are feelings of shame and lack of spontaneity due to the increased bowel activity, and in extreme cases, as in P14, additionally due to the need to wear a diaper:

> "It annoys me a lot because I always have to plan. I can't really spontaneously say, 'Okay, I'm going somewhere now,' because I never know, is it going to be okay now or do I have to go to the toilet. And always having diapers with you or putting them in, that's also very inconvenient in the summer. It is visible." (P14, female, 53 years)

These consequences, reveal very burdening for the patients and can generate additional stress. Furthermore, they perceive that stress in general is strongly related to the occurrence of the disease, its severity and its course in waves, e.g., flare-ups, what leads to insecurities in daily life. Overall, the narratives indicate that as a result the patients' *subjective well-being* [43, 44] and *quality of life* are impaired.

## Various reasons for participating in the program

Most patients have tried several conventional therapies, e.g., pharmacotherapy or even surgery, in the past to find the "appropriate" treatment for their disease. However, from patients' point of view, these therapies did not bring long-lasting relief. In addition, negative *side effects of medications* or a *lack of information and support* in managing the disease from previous treating physicians also encouraged some patients to participate in the program. Overall, patients reported unmet needs. In this case they are *seeking a complement to pure drug treatment of symptoms* and *want a more comprehensive, holistic approach*. They were looking for self-help options as a complement to conventional pharmacotherapy, to actively manage the disease. In this regard some patients aimed *to deepen or supplement their existing knowledge of Complementary and Alternative Medicine (CAM)* by participating in the program, while others were looking for *new approaches and assistance in dealing with their disease*. In this respect, an important aspect is the *reduction of perceived stress* since they themselves perceive a negative effect of stress on the course of the disease. Patients are aware of the fact that they have to continue living with the disease, so they aim for sustainable and long-term intervention effects and improvement:

> "I have always had the feeling: Yes, what else can I do? Because medicine [medication] alone is not enough. I have to go into it on other levels to get a better grip on it [manage it]. And yes, I went to the physician. He doesn't know anything about nutrition and what else you can do, yes, you are left in the dark." (P03, female, 54 years)

## Various perceived changes

In the context of the day clinic intervention, *psychological changes (cognitive/emotional)* have been achieved in the participants. These changes seem to be related to behavioral changes. In this regard *improved mindfulness in everyday life*, especially with regard to stressful situations was reported: "The perception and handling of stress is completely different" (P10, male, 28 years). *Mindfulness*, focus on one's own disease and body, but also specific techniques and exercises learned, help to manage the disease in everyday life. The majority of patients benefit from taking an active part in coping with the disease, which then no longer makes them feel helpless and therefore improves perceived *self-efficacy*:

"The entire perspective on stress is different and, accordingly, the way I deal with it is also different. I'm now consciously aware of it and, of course, I also try to deal with it when I notice that I'm having a very stressful day and that it's bothering me. I'm more aware of it now than I used to be and can of course take appropriate countermeasures." (P10, male, 28 years)

Overall, the narratives show the pattern of *more conscious perception and better assessment* of the disease in everyday life and in dealing with the disease. Through these *cognitive changes*, among other things, a better *acceptance of the disease* could be achieved in some cases. In addition, participants reported that they are considerably more *relaxed and at ease* in everyday life in regard to the disease and symptoms. Some patients report that the *disease has fade into the background*, i.e., no longer has such a high, negatively burdened significance in the patients' lives. P05 sums it up: "[. . .] I nearly feel like a new person. [. . .]" (P05, male, 52 years).

This change was particularly evident in patients who were able to improve their perceived *self-efficacy* and now actively work on to improve their health situation with their newly achieved confidence in dealing with the disease. Furthermore, this was also evident in patients who perceived improvement at the symptomatic level (e.g., less frequent/infrequent bowel movements).

On the *emotional level*, *positive feelings* in everyday life in relation to the disease were reported. Patients talk about being more laid back in everyday life and about *a better emotional balance*. In addition, they have *more energy and/or motivation* and feeling better:

"Yes, I think that a little bit of emotional balance I reached already helps me in that direction. And that I now simply have better strategies to help myself. And I would say that I simply have an awareness of things like setting priorities, taking time for things, being more aware of that. So that has already changed with me, or has definitely remained." (P02, female, 22 years)

A large part of this is due to the newly learned way of dealing with the disease, perceived higher self-efficacy and perceived health improvements. Even with continuing symptoms, they report from a higher stability or a better feeling, higher well-being despite symptoms.

*Changes at the physical/symptomatic* level did not appear in all interviews, but a subset of patients repeatedly reported improvement in symptoms to *greater stability*, i.e., less (abdominal) pain, less diarrhea, fewer bowel movements:

"I used to go to the toilet five or six times when I was in a shopping center. But now that's going quite well at the moment. I haven't experienced it like this for a long time, [. . .] I'm definitely feeling better with the diarrhea. And, yes, it's a much better quality of life now, in any case." (P09, female, 64 years)

The patients relate these improvements to cognitive and emotional changes, e.g., improvement of mindfulness and stress reduction in everyday life. In addition, patients mention *conscious diet and nutrition*, *naturopathic self-help strategies* or the general new developed *holistic view* on their disease in the context of symptom improvements, i.e. as perceived potential reasons for improvements.

Regarding *social changes* some patients report being able *to improve their social life*, *social relations*, *leisure time activities*, *or work* due to better coping strategies. One reason for this is more confidence in dealing with the disease, e.g., meeting with acquaintances without fear of sudden bowel movements or dealing with the disease more calmly at work as well:

"I was always afraid that I would have to go to the toilet or that I wouldn't make it to the toilet. [. . .] Or when you're out and about, where's the next toilet if I have to go. And that, I think, has really changed completely for me. So, I can assess with my body, I think, better, I know when I need to or when I have problems, and otherwise I don't worry about it anymore." (P15, female, 33 years)

In general, *the perceived changes as a result of participation in the program* were evident on different areas and could occur in patients together (e.g., cognitive/emotional and physiological/symptomatic) or could be visible only on one dimension, e.g., only in the perception of the disease. However, one patient reported that a correlation between perceived improvements and participation in the program was difficult to single out for him. The other patients in the sample narrated perceived associations between participation in the program and improvements due to participation.

### Individual and adequate integration of the program elements into everyday life

Overall, the narratives reveal that the patients have tried out and *integrated various elements from the learned repertoire of techniques and program content.* They learned about many options and use them after the program according to the individual course and their personal opportunities. The implementation is very individual, and is determined by the individual's needs, available time resources, physical possibilities as well as personal interests and preferences. They therefore found *adequate techniques for themselves.* Moreover, in this regard, it became apparent that a *new attitude or perspective* on the disease could be gained.

Although the *integration of program elements into everyday life varies greatly in terms of types and extend*, a pattern is evident that all patients have *integrated several elements*; especially those that are easier to integrate into everyday life, i.e., for which either the necessary time can be provided and a routine found and those that take less time and can be used flexibly during the day, e.g., breathing techniques, stress reduction, mindfulness exercises. In almost all interviews it was reported that stress management techniques, i.e., progressive muscle relaxation (PMR), body scan, meditation, relaxation, short breathing exercises were used. The latter, as it can be implemented flexibly and is not tied to a specific location or equipment: "When I simply notice that I'm not feeling well at all, then I do these breathing minis exercises, that also works very quickly [. . .] that also works in between" (P06, female, 37 years).

Additionally, *dietary changes*, or being aware of one's diet, also was relevant in the patients' daily lives. In this respect, the teaching kitchen in the program was very important, as the patients were able to learn about nutrition in a practical way under supervision, and were thus able to take away helpful ideas for their everyday lives:

"Well, one clear thing is actually the diet. There are many things that we now cook and prepare that I would never have done before [. . .] so it's much more nutrition conscious [. . .] And many things have then, I'll say, become habitual or have become part of everyday life, which is something positive." (P01, male, 61 years)

*Exercises* in the form of movement, walking, and Yoga were also relevant and mentioned positively across the interviews and are applied according to one's own interests and physical condition. Further *naturopathic self-help strategies*, such as teas, blueberry juice, bread drink, myrrh, compresses, water applications/hydrotherapy (according to Kneipp), etc., were also

used satisfactorily by some patients after the program, but rather less frequently and intense in relation to the afore mentioned.

Particularly *helpful* for the integration into everyday life was setting fixed appointments, using the downloads and documents provided, and the support from the partner and/or other family members. Most useful mentioned was building up a routine for the applications:

"Well, you have to get a certain routine into it. Yes. And it's difficult to get into that routine at first. As soon as you've got it in, it actually went well [. . .] And I had the feeling that you actually notice when you don't do it, that you're missing it." (P10, male, 28 years)

On the contrary, experienced stress in everyday life, lack of time due to work and/or family commitments, were identified as *obstacles* to implementation. Moreover, a lack of motivation was stated as the greatest hindrance of implementing the program elements: "I would say maybe with some things it's kind of prioritizing and I think for me mainly it's my weaker self" (P02, female, 22 years).

## Overall satisfaction with the program

These perceived positive changes in the majority of patients and the possibility to find and implement helpful applications for oneself in everyday life are also mirrored in the overall satisfaction with the program. Statements about the program *meeting or exceeding expectations* clearly state that:

"It actually exceeded what I had promised myself. [. . .] That you learn something new and you get educated, more enlightened. That's more than a physician actually does." (P12, female, 66 years)

In addition, willingness to participate again, talk about recommendations of the day clinic program (acquaintances, treating physician), etc. indicate how positively the program was experienced by the majority of participants: "And so for me, I must say, it has helped a lot. Thus, I can totally recommend it [the program] to others" (P15, female, 33 years old).

Elements from the program as a complement therapy, such as mindfulness training, stress reduction, relaxation were evaluated particularly positively and beneficially. In addition, the program as a whole with its holistic approach, the exchange of experience with other participants, the mix of practice and theory, and the support and guidance provided during the program were all rated positively:

"I think this day clinic is a great thing. It's a pity I didn't know about it in the past. At that time, I felt very left alone. No physician could help me any further. I got my medication, but this, if I had it back then, it would have been a great enrichment." (P17, female, 55 years)

However, one patient perceived participation as *stressful* and so the intervention was only partially successful for her. Other participants also expressed a kind of *strain/effort and strong time commitment* during participation, due to, family issues or work.

In general, *suggestions for improvement* reflect mainly on personal preferences, but some were mentioned more often across the interviews; for example, intervals between day clinics should be longer, e.g. every two weeks, exercise after lunch was not that pleasant, or some theory sessions were exhausting, while for certain topics longer slots should be provided for more content.

**Fig 2. Theme map–main results.**

In summary, Fig 2 shows the major themes developed during the analyses and gives overview of the main results:

## Discussion

The results of the present interview study indicate that, first, CD can have a very comprehensive and broad impact (psychological, physical, social) on those affected and thus can negatively determine everyday life. As a result, quality of life and well-being can be reduced. Second, there were many and various reasons for participating in the day clinic program. Among other things, patients were looking for self-help options as a complement to conventional pharmacotherapy, to actively manage their disease. Moreover, they did not feel sufficiently supported and informed by their treating physicians. Third, regarding the day clinic

program, the patients were very satisfied with the program itself and experienced relevant support for everyday life in dealing with the disease. They were able to integrate therapy elements that were appropriate for them into their daily lives. Patients report broadly perceived improvements, primarily in the psychological domain, e.g., regarding mindfulness, self-efficacy, emotional balance, etc., but also to some extent in the physical, i.e., symptomatic, and social domains, but these tend to be more individually in cases.

Existing studies–the present empirical study confirms these findings–demonstrate that patients' *quality of life is often impaired* because of IBD symptoms and the associated psychological impact on patients' daily lives [1]. In addition, the present study as well as our previous one with ulcerative colitis patients [28], indicate that there are many IBD patients who do not feel well or sufficiently cared for by their physicians so far and showed unmet needs. Therefore, they were looking for complementary procedures. In addition, other reasons for the use of CAM, such as side effects of conventional medicine/pharmacotherapy were reported [45]. The analysis show that patients are aware of the correlation between perceived stress and disease activity or disease progression, e.g., relapses [23, 45, 46], which can encourage IBD patients using multimodal approaches that include stress reduction as a complement [28, 45, 47].

In this study, from the patients' perspective, through participating in the day clinic program, a *new perspective on the disease* could be achieved, above all through greater mindfulness. This provides new opportunities and confidence in dealing with the disease, i.e., greater perceived self-efficacy. Learning to cope with the disease in general and especially with perceived stressful situations in daily life can play an important role in improving quality of life despite the disease (and its symptoms). Regarding this, previous studies already indicated that therapies and programs that focus mind-body approaches, stress management, mindfulness, meditation, relaxation, etc., can improve the impaired quality of life of IBD patients and, in some cases decrease pain [8, 17–28]. In addition to the numerous program contents and elements applied by the patients, it appears particularly important that patients acquire the necessary skills to help themselves, i.e., to improve their own self-efficacy. In the present study patients implemented stress reduction techniques in particular, because, from their point of view, it can be easily integrated into everyday life (routine) and is helpful.

In addition, referring to the secondary outcomes of the randomized controlled trial of this mixed methods approach [30] anxiety and depression (HADS) decreased significantly for the group of day clinic participants (intervention group) compared to the waiting group (control group) after intervention (week 12). This reduction remained until 9-months follow-up-assessment. Moreover, the quantitative results of the randomized controlled trial showed that 79 percent of the patients in the intervention group compared to 44 percent of the control group improved the disease-related quality of life (IBDQ) at the time of the post intervention measurement (week 12) in a clinically relevant way. At 9-months follow-up-assessment this effect decreased but is still present in about 63 percent of the patients (intervention group). In the post-intervention measurement (week 12), self-efficacy was found to have improved [30], which was also indicated by the qualitative interviews at week 24, as already mentioned, showing a significant improvement in self-efficacy from the patients' perspective. Patients also reported that during the intervention they felt enabled to do something against their disease, that feelings of powerlessness and helplessness had decreased, and that they felt more confident in dealing with their disease. In this regard, the patients reported about the relevance of a routine of the applications. Based on these quantitative and qualitative results, we therefore assume an improvement of self-efficacy.

However, improving the quality of life is not solely achieved through single elements such as stress management and mindfulness in daily life. Here it became clear that it was not individual parts of the overall package that were decisive, but the comprehensive, holistic treatment

approach. The interviews indicate that it is primarily the use of several elements of the multi-modal concept together that is perceived as beneficial for patients. Conscious nutrition is also mentioned in this context. It is quite easy to implement in everyday life and inexpensive. As there are multiple causes or factors in CD, e.g., different lifestyle factors it seems to be advantageous to use multimodal approaches that address lifestyle of patients with IBD [7, 15, 16, 28]. The comprehensive lifestyle modification program underlying this study [48] combines relevant elements of complementary medicine, e.g., mind-body medicine, herbal medicine, exercise, nutrition, and naturopathic self-help strategies. Regarding this combination of theory and practice, guidance by therapists and implementation in everyday life on their own as self-help, proved to be relevant. The present results, as well as existing studies [5, 6, 45] indicate a high complexity and individuality in the manifestation of the disease in the patients. Therefore, a comprehensive, holistic therapy approach can be helpful and also necessary. This was addressed in the day clinic through a multimodal approach in which multiple elements were provided and new perspective on the disease was achieved, giving patients adequate forms of support for everyday life by assisting them to help themselves. The program therefore offers a large repertoire of support options, according to the patient's personal needs. Depending on their current condition, an individual therapy approach can be found that can have a supporting effect in addition to conventional pharmacotherapy. Patients showed very individually perceived changes and improvements in different fields, e.g., emotional, cognitive, but also social and symptomatic are interrelated and, when considered as a whole–not all areas necessarily have to be improved together–finally can lead to an improvement in the quality of life. In addition, the narratives revealed that patients are very glad to have a complementary opportunity to cope with the disease. For some patients, it is even not decisive that physical symptoms could not be improved or could only be partially improved by participation in the day clinic–significant differences in disease activity (HBI) could not be found quantitatively, neither in the intervention group nor in the control group. However, in the intervention group an improvement in inflammatory stool parameters (e.g., fecal CRP and lactoferrin) were found [30].

Although the program received high satisfaction overall, there were some suggestions for improvement. While 86% of the participants completed the entire intervention [29], feedback from the interviews highlighted challenges balancing program participation with employment, causing additional stress. In this context, it is beneficial for participants that the day clinic is covered by the health insurance system now, and that a certificate of incapacity for work is granted for participation in future programs.

Methodological limitations lie in the nature of qualitative research regarding generalization of results. However, relevant patterns for the research interest could be developed. It is important to mention again that there were no patients with acute relapses in the sample, but patients with a moderate course. In addition, social desirability bias during the interviews could not be ruled out completely. However, this risk is considered to be low, since the participants had voluntarily participated in the interview part of the study, the interviews took part after the intervention, and the interviewers were not involved in the previous intervention. Although the results at three months after the end of the intervention, i.e. approximately half a year after the start of the intervention, are positive and promising, the present study does not provide conclusive insights into the long-term effects of the intervention. Further studies are needed in this regard.

## Conclusions

A day clinic program that combines multimodal stress reduction and comprehensive lifestyle modification appears to have positive effects on patients with CD in several ways. Patients

learn about non-pharmaceutical therapeutic options they can use in their daily lives. Patients are empowered to select and use appropriate therapies elements for their individual symptoms and needs. Additionally, they enhance their understanding of the interrelationships of the disease and achieve a more holistic view of the condition. Furthermore, by enhancing mindfulness, self-efficacy, and other related factors, patients experience a more positive everyday life.

The mixed methods research approach proved to be beneficial in understanding the patients' situation and the effects of the intervention through the additional qualitative perspective. Both quantitative and qualitative results indicate that an improvement in quality of life can be achieved. Further research on long-term effects is needed.

## Supporting information

**S1 Fig. Schedule of multimodal stress management and comprehensive lifestyle modification program [30].**
(TIF)

## Acknowledgments

We would like to thank Prof. Dr. Thomas Keil and his team for their support and cooperation in the joint project. We are also very grateful to the German Crohn's and Colitis Association (DCCV e.V.) for their administrative support throughout the recruiting process. We also thank the study patients who dedicated their time and effort to participate in this complex intervention study, the physicians and therapists and Tanja Neufeld and Luise Leithäuser (all Bamberg, Germany) for their assistance and support of the study.

## Author Contributions

**Conceptualization:** Christoph Schlee, Nina Bauer, Jost Langhorst.

**Formal analysis:** Christoph Schlee, Christine Uecker, Özlem Öznur.

**Funding acquisition:** Jost Langhorst.

**Investigation:** Christoph Schlee, Christine Uecker.

**Methodology:** Christoph Schlee, Christine Uecker.

**Project administration:** Nina Bauer, Jost Langhorst.

**Supervision:** Jost Langhorst.

**Writing – original draft:** Christoph Schlee, Christine Uecker, Özlem Öznur.

**Writing – review & editing:** Christoph Schlee, Christine Uecker, Özlem Öznur, Nina Bauer, Jost Langhorst.

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
