## [Decision Letter · Decision Letter 0]

26 Jun 2024

PONE-D-24-13340Participants’ perspectives on a multimodal stress management and comprehensive lifestyle modification program for patients with Crohn’s Disease – a qualitative interview studyPLOS ONE

Dear Dr. JLanghorst,

Thank you for submitting your manuscript to PLOS ONE. After careful consideration, we feel that it has merit but does not fully meet PLOS ONE’s publication criteria as it currently stands. Therefore, we invite you to submit a revised version of the manuscript that addresses the points raised during the review process.

We look forward to receiving your revised manuscript.

Kind regards,

Mehran Rahimlou, PhD

Academic Editor

PLOS ONE

Additional Editor Comments:

Dear Dr. Jost Langhorst

based on the comments of the reviewer, the authors are requested to revise their manuscript based on the comments raised and submit it again after correcting it.

Many thanks

Reviewers' comments:

Reviewer's Responses to Questions

**Comments to the Author**

1. Is the manuscript technically sound, and do the data support the conclusions?

Reviewer #1: Yes

Reviewer #2: Yes

Reviewer #3: Yes

2. Has the statistical analysis been performed appropriately and rigorously? 

Reviewer #1: N/A

Reviewer #2: I Don't Know

Reviewer #3: Yes

3. Have the authors made all data underlying the findings in their manuscript fully available?

Reviewer #1: No

Reviewer #2: Yes

Reviewer #3: Yes

4. Is the manuscript presented in an intelligible fashion and written in standard English?

Reviewer #1: Yes

Reviewer #2: Yes

Reviewer #3: Yes

5. Review Comments to the Author

Reviewer #1: Clinically relevant qualitative study finding that a multimodal stress reduction and lifestyle modification day clinic appears

to be beneficial in the short term as a complementary therapy for CD patients, thereby offering additional options and and helping patients to address individual symptoms and needs, improve their understanding of the disease and their quality of life.

Reviewer #2: The authors described that the interviewers completed the interview protocol as field notes to record any idiosyncrasies or problems encountered during the interviews. These results should also be described if any.

For individual patient, please describe the number and characteristics of the interviewees and their relationship to the patient, as these may affect the results.

The integrated program of stress reduction and lifestyle modification (including, among other things, nutritional therapy and phytotherapy) in the form of a day clinic should be briefly explained also in this manuscript for the readers who would know this intervention for the first time.

Reviewer #3: This study by Schlee et al. examined the efficacy of multimodal integrative approach for patients with Crohn’s disease. I have some comments for this study.

1. "Overall, it is evident that IBD patients suffer from a reduced quality of life, due to

2. symptoms, such as abdominal pain, diarrhea, flatulence, as well as psychosocial symptoms." Add references.

3. "The causes and factors influencing CD are not yet clearly understood. However, it is assumed that, in addition to genetic factors, environmental factors, or lifestyle factors such as perceived stress in everyday life play a major role and impact the course of the disease" there are some other reasons that involved in the etiology of CD such as poor dietary patterns and lack of some nutrients (you can read and add this reference for this sentence: 10.1002/ptr.7081, 10.22037/ghfbb.v16i1.2622, 10.1016/j.dsx.2022.102440)

4. "The study was conducted as a mixed methods approach at the Klinikum Bamberg in" add reference.

5. What is the inclusion and exclusion criteria for choosing the participants.

6. "A total of 37 patients diagnosed with CD were enrolled in the randomized controlled trial" what is the approach in sample size calculation.

7. Whether or not the patients were assigned to the intervention and control group in random form. It should be explained in text.

8.

6. PLOS authors have the option to publish the peer review history of their article (what does this mean?). If published, this will include your full peer review and any attached files.

Reviewer #1: No

Reviewer #2: No

Reviewer #3: No

---

## [Author Response · Author response to Decision Letter 0]

12 Aug 2024

Response to Reviewers for manuscript Schlee et al.

Dear Editor, Dear Reviewers,

Thank you very much for your careful attention to our manuscript and the reviews provided. 

We appreciated the opportunity to revise the manuscript and to respond to the reviews point by point. We have done our best to address all the comments in the best possible way and believe that the paper has benefited from the revisions. We hope that you are satisfied with the changes and find the manuscript appropriate and suitable for publication. Please contact us if you have any questions or concerns.

Review Comments to the Author

Reviewer #1: 

Clinically relevant qualitative study finding that a multimodal stress reduction and lifestyle modification day clinic appears to be beneficial in the short term as a complementary therapy for CD patients, thereby offering additional options and and helping patients to address individual symptoms and needs, improve their understanding of the disease and their quality of life.

Response from the authors: Thank you very much.

Reviewer #2: 

The authors described that the interviewers completed the interview protocol as field notes to record any idiosyncrasies or problems encountered during the interviews. These results should also be described if any.

Response from the authors: Thank you for this comment. There were no problems or peculiarities during the interviews/data collection. We added the sentence: “In this respect, there were no particular incidents or problems that would have affected the quality of the interview” on page 8.

For individual patient, please describe the number and characteristics of the interviewees and their relationship to the patient, as these may affect the results.

Response from the authors: 

Thank you for this important comment. Two sociologists and one M.Sc. in naturopathy and complementary medicine were involved as interviewers from the research team. The interview was conducted by one interviewer each. All interviewers were specialists in conducting qualitative research and were adequately trained in interviewing. The interviewers did not know the people personally, had no previous contact with the patient, and were not involved in the intervention, in order to ensure the greatest possible openness in conducting the interview and not to influence the interview situation in any other way. We have added this information to the manuscript on pages 8 and 9.

The integrated program of stress reduction and lifestyle modification (including, among other things, nutritional therapy and phytotherapy) in the form of a day clinic should be briefly explained also in this manuscript for the readers who would know this intervention for the first time.

Response from the authors: 

Thank you for this hint. The program described in this study is described in more detail on pages 5 and 6. To provide a more complete understanding of the program, we have included additional examples on page 6.

Reviewer #3: This study by Schlee et al. examined the efficacy of multimodal integrative approach for patients with Crohn’s disease. I have some comments for this study.

1. "Overall, it is evident that IBD patients suffer from a reduced quality of life, due to

2. symptoms, such as abdominal pain, diarrhea, flatulence, as well as psychosocial symptoms." Add references.

Response from the authors: 

Thank you for this important comment. We added two references Singh et al. 2011 and Barberio et al. 2021 on page 4. 

3. "The causes and factors influencing CD are not yet clearly understood. However, it is assumed that, in addition to genetic factors, environmental factors, or lifestyle factors such as perceived stress in everyday life play a major role and impact the course of the disease" there are some other reasons that involved in the etiology of CD such as poor dietary patterns and lack of some nutrients (you can read and add this reference for this sentence: 10.1002/ptr.7081, 10.22037/ghfbb.v16i1.2622, 10.1016/j.dsx.2022.102440)

Response from the authors: 

Thank you for the additional literature suggestions and the hint to include dietary/nutritional factors. We have added the reference Morshedzadeh et al for a very good overview article in this respect on page 4.

4. "The study was conducted as a mixed methods approach at the Klinikum Bamberg in" add reference.

Response from the authors: 

We added "Bamberg, Germany" on page 5 to make clear where the study took place. References to the main study/mixed methods research approach are included in the reference [24,25] at the end of the sentence. Please let us know if you mean another specific reference in this context. Thank you.

5. What is the inclusion and exclusion criteria for choosing the participants.

Response from the authors: 

Thank you very much. We have included more detailed information on this topic in addition to the previously mentioned information on page 7.

6. "A total of 37 patients diagnosed with CD were enrolled in the randomized controlled trial" what is the approach in sample size calculation. 

Response from the authors: 

Thank you. As this was a feasibility and pilot study, sample size was not calculated. We included the sentence on page 7: In this respect no sample size calculation was done because this was a feasibility and pilot study

7. Whether or not the patients were assigned to the intervention and control group in random form. It should be explained in text.

Response from the authors: 

Yes, they were randomly assigned to intervention and control group. We included one sentence on this in the manuscript on page 7 to make this clearer. Thank you very much!

---

## [Decision Letter · Decision Letter 1]

11 Sep 2024

PONE-D-24-13340R1Participants’ perspectives on a multimodal stress management and comprehensive lifestyle modification program for patients with Crohn’s Disease – a qualitative interview studyPLOS ONE

Dear Dr. Langhorst,

Thank you for submitting your manuscript to PLOS ONE. After careful consideration, we feel that it has merit but does not fully meet PLOS ONE’s publication criteria as it currently stands. Therefore, we invite you to submit a revised version of the manuscript that addresses the points raised during the review process.

We look forward to receiving your revised manuscript.

Kind regards,

Mehran Rahimlou, PhD

Academic Editor

PLOS ONE

Journal Requirements:

Additional Editor Comments:

Authors are requested to revise their manuscript based on the reviewers' comments.

Reviewers' comments:

Reviewer's Responses to Questions

**Comments to the Author**

1. If the authors have adequately addressed your comments raised in a previous round of review and you feel that this manuscript is now acceptable for publication, you may indicate that here to bypass the “Comments to the Author” section, enter your conflict of interest statement in the “Confidential to Editor” section, and submit your "Accept" recommendation.

Reviewer #1: All comments have been addressed

Reviewer #2: All comments have been addressed

Reviewer #3: (No Response)

2. Is the manuscript technically sound, and do the data support the conclusions?

Reviewer #1: Yes

Reviewer #2: Yes

Reviewer #3: Partly

3. Has the statistical analysis been performed appropriately and rigorously? 

Reviewer #1: N/A

Reviewer #2: N/A

Reviewer #3: Yes

4. Have the authors made all data underlying the findings in their manuscript fully available?

Reviewer #1: Yes

Reviewer #2: Yes

Reviewer #3: No

5. Is the manuscript presented in an intelligible fashion and written in standard English?

Reviewer #1: Yes

Reviewer #2: No

Reviewer #3: Yes

6. Review Comments to the Author

Reviewer #1: I am happy with the revisions made to this paper and believe that it is fit to be published in this journal.

Reviewer #2: (No Response)

Reviewer #3: This study entitled: Participants’ perspectives on a multimodal stress management and comprehensive

lifestyle modification program for patients with Crohn’s Disease – a qualitative interview study is interesting. However, I have some comments.

1. Its better to add a IBD definition at the start of introduction

2. "Poor dietary habits and deficiencies in some nutrients are also implicated in the etiology of IBD" add more reference like this reference 10.1002/ptr.7081

3. "In addition, it is suggested that the predominant dietary habits, e.g., highly processed foods, sugar consumption, etc., in modern societies can have a negative influence" add more reference like 10.1016/j.clinthera.2020.12.021 and 10.1016/j.dsx.2015.09.015

4. At the first of the Method section its better to add the design of this study.

5. How do you calculate sample size?

6. In the method section, its better to add primary and secondary outcomes.

7. The quality of figures are so poor and it need to improve.

7. PLOS authors have the option to publish the peer review history of their article (what does this mean?). If published, this will include your full peer review and any attached files.

Reviewer #1: No

Reviewer #2: No

Reviewer #3: No

---

## [Author Response · Author response to Decision Letter 1]

10 Oct 2024

Response to Reviewers for manuscript Schlee et al.

Dear Editor, Dear Reviewers,

Thank you very much for your careful attention to our manuscript and the reviews provided. We appreciated the opportunity to revise again the manuscript and to respond to the reviewers’ comments. We have done our best to address all the comments to the best of our ability and believe that the paper has again benefited from the revisions. We hope that you are satisfied with the changes and find the manuscript appropriate and suitable for publication. Please contact us if you have any questions or concerns.

Sincerely yours,

Prof. Dr. Jost Langhorst

Review Comments to the Author

Reviewer #1: 

All comments have been addressed.

I am happy with the revisions made to this paper and believe that it is fit to be published in this journal.

Response from the authors: Thank you very much.

Reviewer #2: 

All comments have been addressed.

Response from the authors: Thank you very much.

Reviewer #3: 

This study entitled: Participants’ perspectives on a multimodal stress management and comprehensive lifestyle modification program for patients with Crohn’s Disease – a qualitative interview study is interesting. However, I have some comments.

1. Its better to add a IBD definition at the start of introduction

Response from the authors: 

Thank you. We added a definition on page 4.

2. "Poor dietary habits and deficiencies in some nutrients are also implicated in the etiology of IBD" add more reference like this reference 10.1002/ptr.7081 

Response from the authors: 

Thank you. We added additional references on page 4.

3. "In addition, it is suggested that the predominant dietary habits, e.g., highly processed foods, sugar consumption, etc., in modern societies can have a negative influence" add more reference like 10.1016/j.clinthera.2020.12.021 and 10.1016/j.dsx.2015.09.015

Response from the authors: 

Thank you. We added further references on page 4.

 4. At the first of the Method section its better to add the design of this study.

Response from the authors: 

Thank you for your comment. The study design is described at the beginning of the methods section. In order to make the design more comprehensive and provide the overview earlier (at the beginning of the section), we have moved the first reference to Figure 1 there (see page 5).

5. How do you calculate sample size?

Response from the authors: 

Thank you. We have answered this question during the previous revision and already made changes in the manuscript, see manuscript and previous response. As this was a feasibility and pilot study, sample size was not calculated. We included the sentence on page 7 during the last revision: In this respect no sample size calculation was done because this was a feasibility and pilot study (see page 7).

6. In the method section, its better to add primary and secondary outcomes.

Response from the authors: 

As this paper based on a qualitative study designed to complement the quantitative investigations and objectives, there were no defined primary and secondary outcomes. To make the research interest clearer, we added the following sentence in the manuscript on page 6: More specifically, the following topics were examined: experience of disease in daily life, reasons for participating in the day clinic program, experiences with the program, perceived changes due to the intervention, and integration of program elements in daily life. Thank you.

7. The quality of figures are so poor and it need to improve.

Response from the authors: 

Thank you very much for this hint. We have improved the figures.

---

## [Decision Letter · Decision Letter 2]

21 Oct 2024

Participants’ perspectives on a multimodal stress management and comprehensive lifestyle modification program for patients with Crohn’s disease – a qualitative interview study

PONE-D-24-13340R2

Dear Dr. Jost Langhorst,

We’re pleased to inform you that your manuscript has been judged scientifically suitable for publication and will be formally accepted for publication once it meets all outstanding technical requirements.

Kind regards,

Mehran Rahimlou, PhD

Academic Editor

PLOS ONE

Additional Editor Comments (optional):

Reviewers' comments:

Reviewer's Responses to Questions

**Comments to the Author**

1. If the authors have adequately addressed your comments raised in a previous round of review and you feel that this manuscript is now acceptable for publication, you may indicate that here to bypass the “Comments to the Author” section, enter your conflict of interest statement in the “Confidential to Editor” section, and submit your "Accept" recommendation.

Reviewer #3: All comments have been addressed

2. Is the manuscript technically sound, and do the data support the conclusions?

Reviewer #3: Yes

3. Has the statistical analysis been performed appropriately and rigorously? 

Reviewer #3: Yes

4. Have the authors made all data underlying the findings in their manuscript fully available?

Reviewer #3: Yes

5. Is the manuscript presented in an intelligible fashion and written in standard English?

Reviewer #3: Yes

6. Review Comments to the Author

Reviewer #3: The authors have made all the necessary changes based on the reviewers' comments. Based on the changes made in the new version of the manuscript, this manuscript meets the necessary conditions for acceptance.

7. PLOS authors have the option to publish the peer review history of their article (what does this mean?). If published, this will include your full peer review and any attached files.

Reviewer #3: No

---

## [Editor Report · Acceptance letter]

4 Nov 2024

PONE-D-24-13340R2 

PLOS ONE

Dear Dr. Langhorst, 

I'm pleased to inform you that your manuscript has been deemed suitable for publication in PLOS ONE. Congratulations! Your manuscript is now being handed over to our production team.

Kind regards, 

on behalf of

Dr. Mehran Rahimlou 

Academic Editor

PLOS ONE